# MelGAN: Generative Adversarial Networks for Conditional Waveform Synthesis

**Kundan Kumar**[*]
Lyrebird AI, Mila, University of Montreal
kundan@descript.com

**Rithesh Kumar**[*]
Lyrebird AI
rithesh@descript.com

**Thibault de Boissiere**
Lyrebird AI

**Lucas Gestin**
Lyrebird AI

**Wei Zhen Teoh**
Lyrebird AI

**Jose Sotelo**
Lyrebird AI, Mila

**Alexandre de Brebisson**
Lyrebird AI, Mila

**Yoshua Bengio**
Mila
University of Montreal, CIFAR Program Co-director

**Aaron Courville**
Mila
University of Montreal, CIFAR Fellow

## Abstract

Previous works (Donahue et al., 2018a; Engel et al., 2019a) have found that generating coherent raw audio waveforms with GANs is challenging. In this paper, we show that it is possible to train GANs reliably to generate high quality coherent waveforms by introducing a set of architectural changes and simple training techniques. Subjective evaluation metric (Mean Opinion Score, or MOS) shows the effectiveness of the proposed approach for high quality mel-spectrogram inversion. To establish the generality of the proposed techniques, we show qualitative results of our model in speech synthesis, music domain translation and unconditional music synthesis. We evaluate the various components of the model through ablation studies and suggest a set of guidelines to design general purpose discriminators and generators for conditional sequence synthesis tasks. Our model is non-autoregressive, fully convolutional, with significantly fewer parameters than competing models and generalizes to unseen speakers for mel-spectrogram inversion. Our pytorch implementation runs at more than 100x faster than realtime on GTX 1080Ti GPU and more than 2x faster than real-time on CPU, without any hardware specific optimization tricks.

## 1 Introduction

Modelling raw audio is a particularly challenging problem because of the high temporal resolution of the data (usually atleast 16,000 samples per second) and the presence of structure at different timescales with short and long-term dependencies. Thus, instead of modelling the raw temporal audio directly, most approaches simplify the problem by modelling a lower-resolution representation that can be efficiently computed from the raw temporal signal. Such a representation is typically chosen to be easier to model than raw audio while preserving enough information to allow faithful

---

[*]Equal contribution

inversion back to audio. In the case of speech, aligned linguistic features (Van Den Oord et al., 2016) and mel-spectograms (Shen et al., 2018; Gibiansky et al., 2017) are two commonly used intermediate representations. Audio modelling is thus typically decomposed into two stages. The first models the intermediate representation given text as input. The second transforms the intermediate representation back to audio. In this work, we focus on the latter stage, and choose mel-spectogram as the intermediate representation.[2] Current approaches to mel-spectogram inversion can be categorized into three distinct families: pure signal processing techniques, autoregressive and non-autoregressive neural networks. We describe these three main lines of research in the following paragraphs.

**Pure signal processing approaches** Different signal processing approaches have been explored to find some convenient low-resolution audio representations that can both be easily modelled and efficiently converted back to temporal audio. For example, the Griffin-Lim (Griffin & Lim, 1984) algorithm allows one to efficiently decode an STFT sequence back to the temporal signal at the cost of introducing strong, robotic artifacts as noted in Wang et al. (2017). More sophisticated representations and signal processing techniques have been investigated. For instance, the WORLD vocoder (MORISE et al., 2016) introduces an intermediate representation tailored to speech modelling based on mel-spectogram-like features. The WORLD vocoder is paired with a dedicated signal processing algorithm to map the intermediate representation back to the original audio. It has been successfully used to carry out text-to-speech synthesis, for example in Char2Wav, where WORLD vocoder features are modelled with an attention-based recurrent neural network (Sotelo et al., 2017; Shen et al., 2018; Ping et al., 2017). The main issue with these pure signal processing methods is that the mapping from intermediate features to audio usually introduces noticeable artifacts.

**Autoregressive neural-networks-based models** WaveNet (Van Den Oord et al., 2016) is a fully-convolutional autoregressive sequence model that produces highly realistic speech samples conditioned on linguistic features that are temporally aligned with the raw audio. It is also capable of generating high quality unconditional speech and music samples. SampleRNN (Mehri et al., 2016) is an alternative architecture to perform unconditional waveform generation which explicitly models raw audio at different temporal resolutions using multi-scale recurrent neural networks. WaveRNN (Kalchbrenner et al., 2018) is a faster auto-regressive model based on a simple, single-layer recurrent neural network. WaveRNN introduces various techniques, such as sparsification and subscale generations to further increase synthesis speed. These methods have produced state-of-the-art results in text-to-speech synthesis (Sotelo et al., 2017; Shen et al., 2018; Ping et al., 2017) and other audio generation tasks (Engel et al., 2017). Unfortunately, inference with these models is inherently slow and inefficient because audio samples must be generated sequentially. Thus auto-regressive models are usually not suited for real-time applications.

**Non autoregressive models** Recently, significant efforts have been dedicated to the development of non-autoregressive models to invert low-resolution audio representations. These models are orders of magnitudes faster than their auto-regressive counterparts because they are highly parallelizable and can fully exploit modern deep learning hardware (such as GPUs and TPUs). Two distinct methods have emerged to train such models. 1.) Parallel Wavenet (Oord et al., 2017) and Clarinet (Ping et al., 2018) distill a trained auto-regressive decoder into a flow-based convolutional student model. The student is trained using a probability distillation objective based on the Kulback-Leibler divergence: $KL[P_{\text{student}}||P_{\text{teacher}}]$, along with an additional perceptual loss terms. 2.) WaveGlow (Prenger et al., 2019) is a flow-based generative model based on Glow (Kingma & Dhariwal, 2018). WaveGlow is a very high capacity generative flow consisting of 12 coupling and 12 invertible 1x1 convolutions, with each coupling layer consisting of a stack of 8 layers of dilated convolutions. The authors note that it requires a week of training on 8 GPUs to get good quality results for a single speaker model. While inference is fast on the GPU, the large size of the model makes it impractical for applications with a constrained memory budget.

**GANs for audio** One family of methods that has so far been little explored for audio modelling are generative adversarial networks (GANs) (Goodfellow et al., 2014). GANs have made steady progress in unconditional image generation (Gulrajani et al., 2017; Karras et al., 2017, 2018), image-to-image translation (Isola et al., 2017; Zhu et al., 2017; Wang et al., 2018b) and video-to-video synthesis (Chan et al., 2018; Wang et al., 2018a). Despite their huge success in computer vision, we have not

seen as much progress in using GANs for audio modelling. Engel et al. (2019b) use GANs to generate musical timbre by modelling STFT magnitudes and phase angles instead of modelling raw waveform directly. Neekhara et al. (2019) propose using GANs to learn mappings from mel-spectrogram to simple magnitude spectrogram, which is to be combined with phase estimations to recover raw audio waveform. Yamamoto et al. (2019) use GANs to distill an autoregressive model that generates raw speech audio, however their results show that adversarial loss alone is not sufficient for high quality waveform generation; it requires a KL-divergence based distillation objective as a critical component. To this date, making them work well in this domain has been challenging (Donahue et al., 2018a).

**Main Contributions**

- We introduce MelGAN, a non-autoregressive feed-forward convolutional architecture to perform audio waveform generation in a GAN setup. To the best of our knowledge, this is the first work that successfully trains GANs for raw audio generation without additional distillation or perceptual loss functions while still yielding a high quality text-to-speech synthesis model.

- We show that autoregressive models can be readily replaced with a fast and parallel MelGAN decoder for raw waveform generation through experiments in universal music translation, text-to-speech generation and unconditional music synthesis albeit with slight quality degradation.

- We also show that MelGAN is substantially faster than other mel-spectrogram inversion alternatives. In particular, it is 10 times faster than the fastest available model to date (Prenger et al., 2019) without considerable degradation in audio quality.

## 2 The MelGAN Model

In this section, we describe our generator and discriminator architectures for mel-spectrogram inversion. We describe the core components of the model and discuss modifications to perform unconditional audio synthesis. We compare the proposed model with competing approaches in terms of number of parameters and inference speed on both CPU and GPU. Figure 1 shows the overall architecture.

### 2.1 Generator

**Architecture** Our generator is a fully convolutional feed-forward network with mel-spectrogram $s$ as input and raw waveform $x$ as output. Since the mel-spectrogram (used for all experiments) is at a $256\times$ lower temporal resolution, we use a stack of transposed convolutional layers to upsample the input sequence. Each transposed convolutional layer is followed by a stack of residual blocks with dilated convolutions. Unlike traditional GANs, our generator does not use a global noise vector as input. We noticed in our experiments that there is little perceptual difference in the generated waveforms when additional noise is fed to the generator. This is a counter-intuitive result because the inversion of $s \rightarrow x$ involves a one-to-many mapping since $s$ is a lossy-compression of $x$. However, this finding is consistent with Mathieu et al. (2015) and Isola et al. (2017), which show that noise input is not important if the conditioning information is very strong.

**Induced receptive field** In convolutional neural network based generators for images, there is an inductive bias that pixels which are spatially close-by are correlated because of high overlap among their induced receptive fields. We design our generator architecture to put an inductive bias that there is long range correlation among the audio timesteps. We added residual blocks with dilations after each upsampling layer, so that temporally far output activations of each subsequent layer has significant overlapping inputs. Receptive field of a stack of dilated convolution layers increases exponentially with the number of layers. Similar to Van Den Oord et al. (2016), incorporating these in our generator allows us to efficiently increase the induced receptive fields of each output time-step. This effectively implies larger overlap in the induced receptive field of far apart time-steps, leading to better long range correlation.

**Checkerboard artifacts** As noticed in Odena et al. (2016), deconvolutional generators are susceptible to generating "checkerboard" patterns if the kernel-size and stride of the transposed convolutional

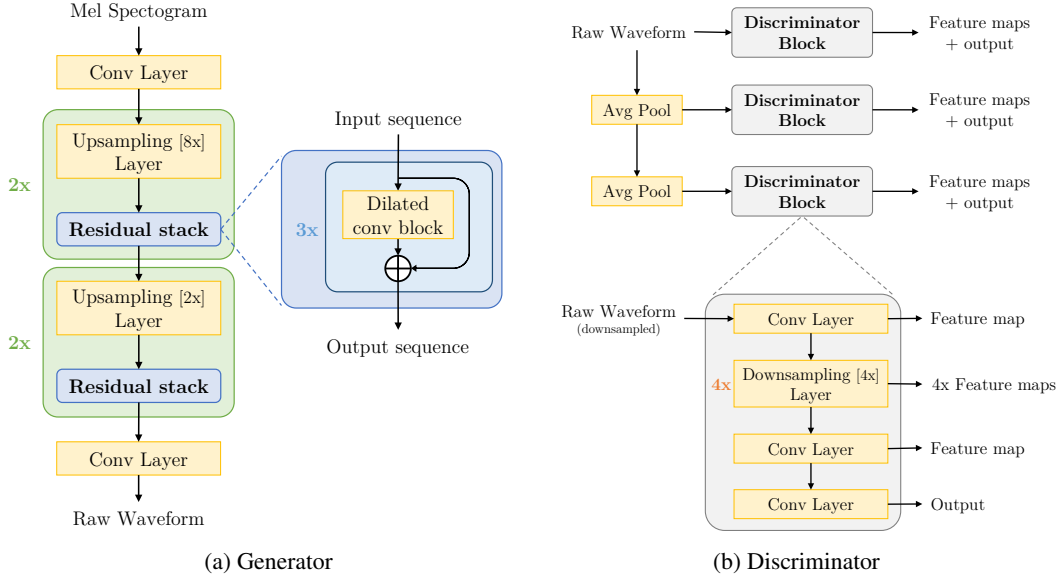

(a) Generator        (b) Discriminator

Figure 1: MelGAN model architecture. Each upsampling layer is a transposed convolution with kernel-size being twice of the stride (which is same as the upsampling ratio for the layer). 256x upsampling is done in 4 stages of 8x, 8x, 2x and 2x upsampling. Each residual dilated convolution stack has three layers with dilation 1, 3 and 9 with kernel-size 3, having a total receptive field of 27 timesteps. We use leaky-relu for activation. Each discriminator block has 4 strided convolution with stride 4. Further details can be found in the Appendix 6.

layers are not carefully chosen. Donahue et al. (2018b) examines this for raw waveform generation and finds that such repeated patterns lead to audible high frequency hissing noise. We solve this problem by carefully choosing the kernel-size and stride for our deconvolutional layers as a simpler alternative to PhaseShuffle layer introduced in Donahue et al. (2018b). Following Odena et al. (2016), we use kernel-size as a multiple of stride. Another source of such repeated patterns, can be the dilated convolution stack if dilation and kernel size are not chosen correctly. We make sure that the dilation grows as a power of the kernel-size such that the receptive field of the stack looks like a fully balanced (seeing input uniformly) and symmetric tree with kernel-size as the branching factor.

**Normalization technique** We noticed that the choice of normalization technique for the generator was extremely crucial for sample quality. Popular conditional GAN architectures for image generation (Isola et al., 2017; Wang et al., 2018b) use instance normalization (Ulyanov et al., 2016) in all the layers of the generator. However, in the case of audio generation we found that instance normalization washes away important important pitch information, making the audio sound metallic. We also obtained poor results when applying spectral normalization (Miyato et al., 2018) on the generator as suggested in Zhang et al. (2018); Park et al. (2019). We believe that the strong Lipshitz constraint on the discriminator impacts the feature matching objective (explained in Section 3.2) used to train the generator. *Weight normalization* (Salimans & Kingma, 2016) worked best out of all the available normalization techniques since it does not limit the capacity of the discriminator or normalize the activations. It simply reparameterizes the weight matrices by decoupling the scale of the weight vector from the direction, to have better training dynamics. We therefore use weight normalization in all layers of the generator.

## 2.2 Discriminator

**Multi-Scale Architecture** Following Wang et al. (2018b), we adopt a multi-scale architecture with 3 discriminators ($D_1, D_2, D_3$) that have identical network structure but operate on different audio scales. $D_1$ operates on the scale of raw audio, whereas $D_2, D_3$ operate on raw audio downsampled by a factor of 2 and 4 respectively. The downsampling is performed using strided average pooling with kernel size 4. Multiple discriminators at different scales are motivated from the fact that audio has structure at different levels. This structure has an inductive bias that each discriminator learns

features for different frequency range of the audio. For example, the discriminator operating on downsampled audio, does not have access to high frequency component, hence, it is biased to learn discriminative features based on low frequency components only.

**Window-based objective**    Each individual discriminator is a Markovian window-based discriminator (analogues to image patches, Isola et al. (2017)) consisting of a sequence of strided convolutional layers with large kernel size. We utilize grouped convolutions to allow the use of larger kernel sizes while keeping number of parameters small. While a standard GAN discriminator learns to classify between distributions of entire audio sequences, window-based discriminator learns to classify between distribution of small audio chunks. Since the discriminator loss is computed over the overlapping windows where each window is very large (equal to the receptive field of the discriminator), the MelGAN model learns to maintain coherence across patches. We chose window-based discriminators since they have been shown to capture essential high frequency structure, require fewer parameters, run faster and can be applied to variable length audio sequences. Similar to the generator, we use weight normalization in all layers of the discriminator.

### 2.3   Training objective

To train the GAN, we use the hinge loss version of the GAN objective (Lim & Ye, 2017; Miyato et al., 2018). We also experimented with the least-squares (LSGAN) formulation (Mao et al., 2017) and noticed slight improvements with the hinge version.

$$\min_{D_k} \mathbb{E}_x \Big[ \min(0, 1 - D_k(x)) \Big] + \mathbb{E}_{s,z} \Big[ \min(0, 1 + D_k(G(s, z))) \Big], \ \forall k = 1, 2, 3 \tag{1}$$

$$\min_{G} \mathbb{E}_{s,z} \left[ \sum_{k=1,2,3} -D_k(G(s, z)) \right] \tag{2}$$

where $x$ represents the raw waveform, $s$ represents the conditioning information (eg. mel-spectrogram) and $z$ represents the gaussian noise vector.

**Feature Matching**    In addition to the discriminator's signal, we use a feature matching objective (Larsen et al., 2015) to train the generator. This objective minimizes the L1 distance between the discriminator feature maps of real and synthetic audio. Intuitively, this can be seen as a learned similarity metric, where a discriminator learns a feature space that discriminates the fake data from real data. It is worth noting that we do not use any loss in the raw audio space. This is counter to other conditional GANs (Isola et al., 2017) where L1 loss is used to match conditionally generated images and their corresponding ground-truths, to enforce global coherence. In fact, in our case adding L1 loss in audio space introduces audible noise that hurts audio quality.

$$\mathcal{L}_{\text{FM}}(G, D_k) = \mathbb{E}_{x, s \sim p_{\text{data}}} \left[ \sum_{i=1}^{T} \frac{1}{N_i} ||D_k^{(i)}(x) - D_k^{(i)}(G(s))||_1 \right] \tag{3}$$

For simplicity of notation, $D_k^{(i)}$ represents the $i^{\text{th}}$ layer feature map output of the $k^{\text{th}}$ discriminator block, $N_i$ denotes the number of units in each layer. Feature matching is similar to the perceptual loss (Dosovitskiy & Brox, 2016; Gatys et al., 2016; Johnson et al., 2016). In our work, we use feature matching at each intermediate layer of all discriminator blocks.

We use the following final objective to train the generator, with $\lambda = 10$ as in (Wang et al., 2018b):

$$\min_{G} \left( \mathbb{E}_{s,z} \left[ \sum_{k=1,2,3} -D_k(G(s, z)) \right] + \lambda \sum_{k=1}^{3} \mathcal{L}_{\text{FM}}(G, D_k) \right) \tag{4}$$

### 2.4   Number of parameters and inference speed

The inductive biases incorporated in our architecture make the overall model significantly smaller than competing models in terms of number of parameters. Being non-autoregressive and fully convolutional, our model is very fast at inference time, capable of running at a frequency of 2500kHz

on GTX1080 Ti GPU in full precision (more than $10\times$ faster than the fastest competing model), and 50kHz on CPU (more than $25\times$ faster than the fastest competing model). We believe that our model is also well-suited for hardware specific inference optimization (such as half precision on Tesla V100 (Jia et al., 2018; Dosovitskiy & Brox, 2016) and quantization (as done in Arik et al. (2017)) which will further boost inference speed. Table 1 shows the detailed comparison.

Table 1: Comparison of the number of parameters and the inference speed. Speed of $n$ kHz means that the model can generate $n \times 1000$ raw audio samples per second. All models are benchmarked using the same hardware [3].

| Model | Number of parameters (in millions) | Speed on CPU (in kHz) | Speed on GPU (in kHz) |
|---|---|---|---|
| Wavenet (Shen et al., 2018) | 24.7 | 0.0627 | 0.0787 |
| Clarinet (Ping et al., 2018) | 10.0 | 1.96 | 221 |
| WaveGlow (Prenger et al., 2019) | 87.9 | 1.58 | 223 |
| **MelGAN (ours)** | **4.26** | **51.9** | **2500** |

## 3 Results

To encourage reproduciblity, we attach the code[4] accompanying the paper.

### 3.1 Ground truth mel-spectrogram inversion

**Ablation study** First, in order to understand the importance of various components of our proposed model, we perform qualitative and quantitative analysis of the reconstructed audio for the mel-spectrogram inversion task. We remove certain key architectural decisions and evaluate the audio quality using the test set. Table 2 shows the mean opinion score of audio quality as assessed via human listening tests. Each model is trained for $400k$ iterations on the LJ Speech dataset (Ito, 2017). Our analysis leads to the following conclusions: Absence of **dilated convolutional stacks** in the generator or removing **weight normalization** lead to high frequency artifacts. Using a **single discriminator** (instead of multi-scale discriminator) produces metallic audio, especially while the speaker is breathing. Moreover, on our internal 6 clean speakers dataset, we notice that this version of the model skips certain voiced portions, completely missing some words. Using **spectral normalization** or removing the **window-based discriminator loss** makes it harder to learn sharp high frequency patterns, causing samples to sound significantly noisy. Adding an extra **L1 penalty** between real and generated raw waveform makes samples sound metallic with additional high frequency artifacts.

Table 2: Mean Opinion Score of ablation studies. To evaluate the biases induced by each component, we remove them one at a time, and train the model for 500 epochs each. Evaluation protocol/details can be found in appendix B.

| Model | MOS | 95% CI |
|---|---|---|
| w/ Spectral Normalization | 1.33 | $\pm 0.07$ |
| w/ L1 loss (audio space) | 2.59 | $\pm 0.11$ |
| w/o Window-based Discriminator | 2.29 | $\pm 0.10$ |
| w/o Dilated Convolutions | 2.60 | $\pm 0.10$ |
| w/o Multi-scale Discriminator | 2.93 | $\pm 0.11$ |
| w/o Weight Normalization | 3.03 | $\pm 0.10$ |
| Baseline (MelGAN) | **3.09** | **$\pm$ 0.11** |

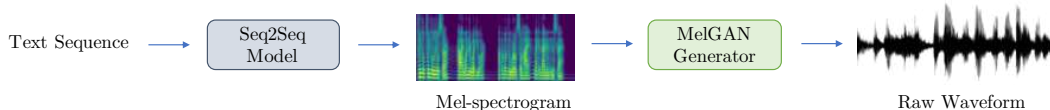

Figure 2: Text-to-speech pipeline.

**Benchmarking competing models**  Next, in order to compare the performance of MelGAN for inverting ground truth mel-spectrograms to raw audio against existing methods such as WaveNet vocoder, WaveGlow, Griffin-Lim and Ground Truth audio, we run an independent MOS test where the MelGAN model is trained until convergence (around $2.5M$ iterations). Similar to the ablation study, these comparisons are made on models trained on the LJ Speech Datset. The results of this comparison are shown in Table 3.

Table 3: Mean Opinion Scores

| Model | MOS | 95% CI |
|---|---|---|
| Griffin Lim | 1.57 | ±0.04 |
| WaveGlow | 4.11 | ±0.05 |
| WaveNet | 4.05 | ±0.05 |
| MelGAN | 3.61 | ±0.06 |
| Original | **4.52** | **± 0.04** |

This experiment result indicates that MelGAN is comparable in quality to state-of-the-art high capacity WaveNet-based models such as WaveNet and WaveGlow. We believe that the performance gap can be quickly bridged in the future by further exploring this direction of using GANs for audio synthesis.

**Generalization to unseen speakers**  Interestingly, we noticed that when we train MelGAN on a dataset containing multiple speakers (internal 6 speaker dataset consisting of 3 male and 3 female speakers with roughly 10 hours per speaker), the resulting model is able to generalize to completely new (unseen) speakers outside the train set. This experiment verifies that MelGAN is able to learn a speaker-invariant mapping of mel spectrograms to raw waveforms.

Table 4: Mean Opinion Scores on the VCTK dataset (Veaux et al., 2017).

| Model | MOS | 95% CI |
|---|---|---|
| Griffin Lim | 1.72 | ±0.07 |
| MelGAN | 3.49 | ±0.09 |
| Original | **4.19** | **± 0.08** |

In an attempt to provide an easily comparable metric to systematically evaluate this generalization (for current and future work), we run an MOS hearing test for ground-truth mel-spectrogram inversion on the public available VCTK dataset (Veaux et al., 2017). The results of this test are shown in Table 4.

### 3.2  End-to-end speech synthesis

We perform quantitative and qualitative comparisons between our proposed MelGAN vs competing models on mel-spectrogram inversion for end-to-end speech synthesis. We plug the MelGAN model in an end-to-end speech synthesis pipeline (Figure 2) and evaluate the text-to-speech sample quality with competing models.

Specifically, we compare the sample quality when using MelGAN for spectrogram inversion vs WaveGlow using Text2mel - an improved version of the open-source char2wav model (Sotelo et al., 2017). Text2mel generates mel-spectrograms instead of vocoder frames, uses phonemes as the input representation and can be coupled with WaveGlow or MelGAN to invert the generated mel-spectrograms. We use this model since its sampler, trains faster and does not perform any mel-frequency clipping like Tacotron2. Additionally, we also include the state-of-the-art Tacotron2 model (Shen et al., 2018) coupled with WaveGlow for baseline comparison. We use the open source implementations of Tacotron2 and WaveGlow provided by NVIDIA in the Pytorch Hub repository to generate the samples. When using WaveGlow, we use the Denoiser with strength $0.01$ provided in the official repository to remove high frequency artifacts. The results of the MOS tests are shown in the table 5.

For all experiments, MelGAN was trained with batch size 16 on a single NVIDIA RTX2080Ti GPU. We use Adam as the optimizer with learning rate of 1e-4 with $\beta_1 = 0.5$ and $\beta_2 = 0.9$ for the

Table 5: Mean Opinion Score of end to end speech synthesis. Evaluation protocol/details can be found in appendix B.

| Model | MOS | 95% CI |
|---|---|---|
| Tacotron2 + WaveGlow | 3.52 | ±0.04 |
| Text2mel + WaveGlow | 4.10 | ±0.03 |
| Text2mel + MelGAN | 3.72 | ±0.04 |
| Text2mel + Griffin-Lim | 1.43 | ±0.04 |
| Original | 4.46 | ± 0.04 |

generator and the discriminators. Samples for qualitative analysis can be found on the accompanied web-page [5]. You can try the speech correction application here [6] created based on the end-to-end speech synthesis pipeline described above.

The results indicate that MelGAN is comparable to some of the best performing models to date as a vocoder component of TTS pipeline. To the best of our ability we also created a TTS model with Text2mel + WaveNet vocoder to add to our comparison. We use the pretrained WaveNet vocoder model provided by Yamamoto (2019) and train the Text2mel model with the required data preprocessing performed. However the model only obtained an MOS score of $3.40 \pm 0.04$. We suspect that

### 3.3 Non autoregressive decoder for music translation

To show that MelGAN is robust and can be plugged into any setup that currently operates using an autoregressive model to perform waveform synthesis, we replace the wavenet-type autoregressive decoder in the Universal Music Translation Network (Mor et al., 2019) with a MelGAN generator.

In this experiment, we use the pre-trained universal music encoder provided by the authors to transform 16kHz raw audio into a latent code sequence of 64 channels, with a downsampling factor of 800 in the time dimension. This implies a $12.5\times$ information compression rate in this domain independent latent representation. Using only the data from a target musical domain, our MelGAN decoder is trained to reconstruct raw waveform from latent code sequence in the GAN setup we described earlier. We adjust the model hyperparameters to obtain upsampling factors of 10, 10, 2, 2, 2 to reach the input resolution. For each selected domain on MusicNet (Thickstun et al., 2018), a decoder is trained for 4 days on an RTX2080 Ti GPU on the available data.

The music translation network augmented with MelGAN decoder is able to perform music translation from any musical domain to the target domain it is trained on with decent quality. We compare qualitative samples from our model against the original model here [5]. The augmented model requires only around 160 milliseconds to translate 1 second of input music audio on an RTX2080 Ti GPU, about 2500 times faster than the original model on the same hardware.

### 3.4 Non-autoregressive decoder for VQ-VAE

Further establishing the generality of our approach, we substitute the decoder in Vector-Quantized VAEs (van den Oord et al., 2017) with our proposed adversarially learned decoder. VQ-VAE is a variational autoencoder which produces a downsampled discrete latent encoding of the input. VQ-VAE uses a high-capacity autoregressive wavenet decoder to learn the data conditional $p(x|z_q)$.

Figure 3 shows an adapted version of VQ-VAE for the task of music generation. In our variant, we use two encoders. The local encoder encodes the audio sequence into a $64\times$ downsampled time series $z_e$. Each vector in this sequence is then mapped to 1 of 512 quantized vectors using a codebook. This follows the same structure as proposed in (van den Oord et al., 2017). The second encoder outputs a global continuous-valued latent vector $y$.

We show qualitative samples of unconditional piano music generation following (Dieleman et al., 2018), where we learn a single tier VQVAE on a raw audio scale, and use a vanilla autoregressive

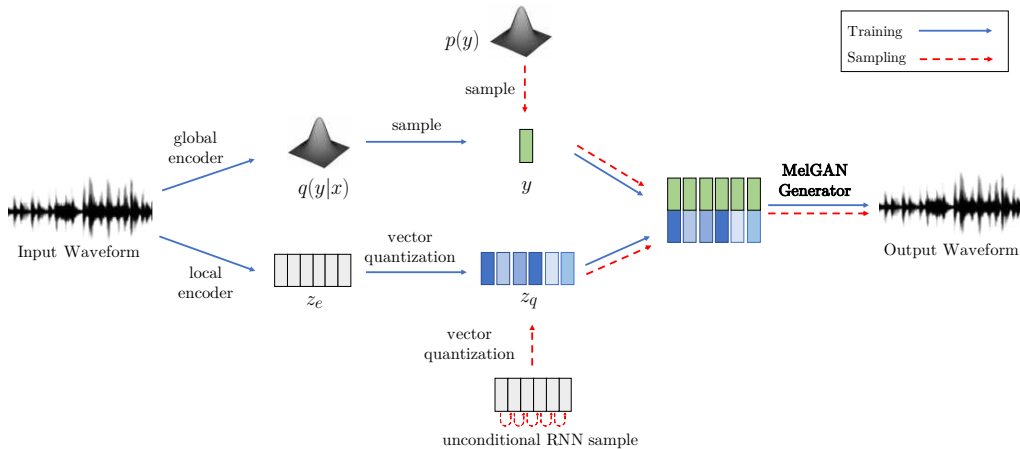

Figure 3: Adapted VQ-VAE model for unconditional music generation. During training, the local encoder downsamples the input information along the time dimension into a sequence $z_e$, which are mapped to a dictionary of vector embeddings to form $z_q$. The global encoder path is the feed-forward path of a vanilla VAE model with gaussian posterior.

.

model (4-layer LSTM with 1024 units) to learn the prior over the sequence of discrete latents. We sample $z_q$ unconditionally using the trained recurrent prior model, and $y$ from a unit gaussian distribution. Qualitatively, conditioned on the same sequence of discrete latents, sampling from the global latent's prior distribution results in low level waveform variations such as phase shifts, but perceptually the outputs sound very similar. We find that the global latent is essential to improve reconstruction quality since it better captures stochasticity in the data conditional $p(x|z_q, y)$, as the discrete latent information learned via the local encoder ($z_q$) is highly compressed. We use latent vector of size 256 and use the same hyper-parameters for training as mel-spectrogram inversion experiment. We used upsampling layers with 4x, 4x, 2x and 2x ratios to achieve 64x upsampling.

# 4    Conclusion and future work

We have introduced a GAN architecture tailored for conditional audio synthesis and we demonstrated qualitative and quantitative results establishing the effectiveness and generality of the proposed methods. Our model has the following assets: it is very lightweight, can be trained quickly on a single desktop GPU, and it is very fast at inference time. We hope that our generator can be a plug-and-play replacement to compute-heavy alternatives in any higher-level audio related tasks.

While the proposed model is well-suited to the task of training and generating sequences of varying length, it is limited by the requirement of time-aligned conditioning information. Indeed it has been designed to operate in the case where the output sequence length is a factor of the input sequence length, which is not always the case in practice. Likewise, feature matching with paired ground truth data is limiting because it is infeasible in some scenarios. For unconditional synthesis, the proposed model needs to defer learning a sequence of conditioning variables to other, better-suited methods such as VQ-VAE. Learning high quality unconditional GAN for audio is a very interesting direction for future work, which we believe will benefit from incorporating the specific architectural choices introduced in this work.

# 5    Acknowledgements

The authors would like to thank NSERC, Canada CIFAR AI Chairs, Canada Research Chairs and IVADO for funding.

## Footnotes

[2]Our methods can likely be used with other representations but this is beyond the scope of this paper.

[3] We use *NVIDIA GTX 1080Ti* for the GPU benchmark and *Intel(R) Core(TM) i9-7920X CPU @ 2.90GHz* processor for the CPU benchmark, tested on only 1 CPU core. We set OMP_NUM_THREADS=1 and MKL_NUM_THREADS=1

[4] `https://github.com/descriptinc/melgan-neurips`.

[5]https://melgan-neurips.github.io

[6]https://www.descript.com/overdub

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
