[Supplementary Material]

# Appendix A   Model Architecture

Table 4: Generator and Discriminator architecture for the mel spectrogram inversion task

| 7 × 1, stride=1 conv 512 |
|---|
| lReLU 16 × 1, stride=8 conv transpose 256 |
| Residual Stack 256 |
| lReLU 16 × 1, stride=8 conv transpose 128 |
| Residual Stack 128 |
| lReLU 4 × 1, stride=2 conv transpose 64 |
| Residual Stack 64 |
| lReLU 4 × 1, stride=2 conv transpose 32 |
| Residual Stack 32 |
| lReLU 7 × 1, stride=1 conv 1 Tanh |

(a) Generator Architecture

| 15 × 1, stride=1 conv 16 lReLU |
|---|
| 41 × 1, stride=4 groups=4 conv 64 lReLU |
| 41 × 1, stride=4 groups=16 conv 256 lReLU |
| 41 × 1, stride=4 groups=64 conv 1024 lReLU |
| 41 × 1, stride=4 groups=256 conv 1024 lReLU |
| 5 × 1, stride=1 conv 1024 lReLU |
| 3 × 1, stride=1 conv 1 |

(b) Discriminator Block Architecture

Figure 4: Residual Stack Architecture

# Appendix B  Hyper-parameters and Training Details

We used a batch-size of 16 for all experiments. Adam, with learning rate 0.0001, $\beta_1 = 0.5$ and $\beta_2 = 0.9$ was used as the optimizer for both generator and discriminator. We used 10 as the coefficient for th feature-matching loss term. We used pytorch to implement our model, and its source code is accompanied with this submission. For VQGAN experiments, we used global latent vector of size 256, with KL term clamped below by 1.0 to avoid posterior collapse. We trained our models on Nvidia GTX1080Ti or GTX 2080Ti. In the supplementary material, we show reconstruction samples as a function of total number of epochs and wall-clock time. We find that our model starts producing intelligible samples very early in training.

# Appendix C  Evaluation Method - MOS

We conducted Mean Opinion Score (MOS) tests to compare the performance of our model to competing architectures. We built the test by gathering samples generated by the different models as well as some original samples. All the generated samples were not seen during training. The MOS scores were computed on a population of 200 individuals: each of them was asked to blindly evaluate a subset of 15 samples taken randomly from this pool of samples by scoring samples from 1 to 5. The samples were presented and rated one at a time by the testers. The tests were crowdsourced using Amazon Mechanical Turk and we required the testers to wear headphones and be English speakers. After gathering all the evaluations, the MOS score $\mu_i$ of model $i$ is estimated by averaging the scores $m_{i,\cdot}$ of the samples coming from the different models. In addition, we compute the 95% confidence intervals for the scores. $\hat{\sigma}_i$ being the standard deviation of the scores collected.

$$\hat{\mu}_i = \frac{1}{N_i} \sum_{k=1}^{N_i} m_{i,k}$$

$$CI_i = \left[ \hat{\mu}_i - 1.96 \frac{\hat{\sigma}_i}{\sqrt{N_i}}, \hat{\mu}_i + 1.96 \frac{\hat{\sigma}_i}{\sqrt{N_i}} \right]$$