[Reviews · NeurIPS 2019]

Reviewer 1



Originality: The first work that trains GAN for spectrogram-to-waveform conversion without distillation. Quality: This paper suffers from a few critical issues. See comments below. Clarity: The experiment setting ups can be described with more details. Sec 3.2 and 3.4 is missing important information such as the datasets used for conducting the experiments. Significance: Although the quality of the proposed model remains unclear because of the previously mentioned critical issues, it’s a significant work because it’s the first GAN-based model for spectrogram-to-waveform conversion which seems to be working at some degree. Critical issues: 1. It’s significantly over-claimed: 1) claiming state-of-the-art for spectrogram-to-waveform conversion (line 6) with MOS 3.09 is surprising; many previous works are at a much higher level (e.g. 3.96 on the same dataset in https://arxiv.org/pdf/1811.00002.pdf, 4.5 in https://arxiv.org/pdf/1712.05884.pdf); 2) claiming state-of-the-art for text-to-speech synthesis (line 87) without comparison to strong baselines; 3) claiming “autoregressive models can be readily replaced with MelGAN decoder” (line 89, line 228) without necessary experiments (e.g. for TTS, no comparison to WaveNet and WaveRNN, no quantitative evaluation for multispeaker TTS; missing strong baselines) ; 4) claiming likely to work for linguistic-feature-to-waveform generation (footnote 1) without evidence. 2. Some experiment results seem in conflict. The MOS scores in a TTS setting up (3.88 in Table 3) is way higher than the MOS scores in a much easier (groundtruth-)spectrogram-to-waveform conversion setting up (3.09 in Table 2). Other comments: 1. Table 2: it will be very helpful to include the MOS for groundtruth audio, which serves as an anchor for comparison MOS scores (which is a subjective evaluation score). 2. Table 3: needs stronger baselines, such as Tacotron + WaveNet and Text-to-Mel + WaveNet, especially because the paper is claiming state-of-the-art. Also needs MOS on groundtruth. 3. It’s not clear what datasets are used for experiments in Sec 3.2 and 3.4. It will be helpful to make it clear. 4. Sec. 3.2: are the vocoder models trained on groundtruth spectrogram or predicted spectrogram? For best results, they should be trained on predicted spectrogram (see https://arxiv.org/pdf/1712.05884.pdf). 5. Since spectrogram-to-waveform conversion is a strongly conditioned generative process, it’s interesting to know how much benefit GAN brings in this work. A baseline to compare with is to train only a Generator model with MSE loss (or other simple loss), without using Discriminator. 6. The author seems misunderstanding what VQ-VAE refers to (Sec. 3.4). VQ-VAE is a framework for learning discrete latent representation, which doesn’t include a WaveNet decoder (line 248) or other decoder (it’s typically paired with a decoder). It doesn't have to produce “downsampled” encoding either (line 247). 7. Line 93 “10 times faster than the fastest available model to date” -- please add a reference to the “fastest available model to date” in order to be clear which model is compared to. 8. Two papers should be covered as related work in the paragraph on line 73: https://arxiv.org/pdf/1904.07944.pdf and https://arxiv.org/pdf/1902.08710.pdf. ============== Update: Thanks for the authors' response. The critical issues were addressed. I updated my score accordingly. However, the authors' response didn't address 6, 7, 8 in my detailed review. Please address them in the camera ready version. A couple more comments regarding the response: 1. Discrepancy in MOS scores between Table 3 and Table 2: comparison before training is converged is generally not trustworthy. Please justify why it's a proper comparison in the camera ready version. 2. The complete failure on the training without Discriminator sounds surprising. It may suggest headroom for improving in the design of the (strongly-conditional) Generator.

Reviewer 2



1. This paper successfully applied GAN objective on neural vocoder for speech synthesis. The model is moderately novel. 2. The paper present the model in a clear way. Equation/figures are accurate. Overall writing is high quality 3. The experimental results are missing the MOS of existing vocoders. For example, the authors should at least provide the MOS of autoregressive wavenet. For copy synthesis, the MOS of the proposed vocoder is only 3.09. The current state-of-the-art is above 4.0. So the authors can not claim the proposed vocoder as state-of-the-art. 4. Copy synthesized speech from the proposed vocoder contains clear artifacts. I don't think this vocoder can replace wavenet or wavernn.

Reviewer 3



UPDATE: having read the other reviews and the authors' rebuttal, I have decided not to change the review score. ---------- This paper proposes an adversarial model for mel-spectrogram inversion. This is an essential step in many modern audio generation pipelines. Mel-spectrogram inversion is challenging because this representation typically contains little to no phase information, and this information is essential to produce realistic sounding audio. While this has traditionally been tackled using signal processing methods (e.g. the Griffin-Lim phase reconstruction algorithm), recently several approaches using generative models have been proposed (e.g. using autoregressive models or flow-based models). The proposed use of an adversarial model in this setting is interesting: it is well known that adversarial models tend to forgo modelling all variations in the data in favour of focusing on a few modes, in order to be able to produce realistic examples given their limited capacity. This is actually a desirable property in the setting of e.g. text-to-speech, where we are simply after realistic conditional speech generation, and we don't care about capturing every possible variation in the speech signal corresponding to a given mel-spectrogram. However, success with adversarial models in the audio domain has been limited so far, and most of the literature has relied on likelihood-based models, so I think this is a timely paper. While the provided recordings of reconstructed and generated speech contain some audible artifacts (short hiccups / "doublespeak" which is characteristic of spectrogram inversion), the results are nevertheless impressive. My intuition is that they would need to be slightly better to be on par with production-level models such as parallel WaveNet, but they are remarkably close. The results in table 3 are impressive as well. The fidelity of the music samples is not so great, however. Remarks: - line 19: note that there is no intrinsic requirement for audio to have a sample rate of at least 16 kHz -- this rate just happens to be used quite commonly in literature for speech signals, because it is high enough for speech generation at a reasonable perceptual quality. - line 36: a very recent paper by Vasquez and Lewis ("MelNet: A Generative Model for Audio in the Frequency Domain") addresses the issue of robotic artifacts by using very high-resolution mel-spectrograms. Although this work wasn't publicly available at the time of submission, and is largely orthogonal in terms of its goals, I think it warrants a mention in the camera ready version of this work. - line 49: while it is true that WaveNet and other autoregressive models have trouble modelling information at time-scales larger than a few seconds, this does not seem particularly relevant in the context of mel-spectrogram inversion, and the criticism could arguably apply to almost any other model discussed in the paper. - Related to the previous comment: in some places, the paper would benefit from a bit more clarity with regards to which task is being considered: mel-spectrogram inversion, or audio generation in a general sense. Some statements only make sense in one context or the other, but it isn't always clear which context is meant. - lines 146-154: the motivation for the multiscale architecture refers to audio having structure at different levels. These "levels" are typically understood to be more than a factor of 2 or 4 apart though, so this motivation feels a bit out of place here. I think a link to wavelet-based processing of audio could perhaps be more appropriate here. - For the comparison in Table 1, it isn't clear at all whether the same hardware was used -- could you clarify? If not, these numbers would be considerably less meaningful, so this needs to be stated clearly. - Table 2 is excellent and clearly demonstrates the impact of the most important model design decisions.

[Author Response · NeurIPS 2019]

We thank all the reviewers for their valuable comments and acknowledging the significance and timeliness of this work.
The reviewers agree that MelGAN is the first GAN-based method for conditional raw waveform synthesis without
distillation or domain specific loss terms. MelGAN has important qualities such as: 1.) fast inference speed (2500 KHz)
2.) trained from scratch and does not require KL-distillation from trained autoregressive models 3.) shows generalization
to unseen speakers for the task of mel-spectrogram inversion, 4.) generalizes to at least three different tasks involving
strongly conditional waveform synthesis. We believe that these important contributions warrant publication in the
conference. To the best of our abilities, we address the following critical comments raised by the reviewers:

**Datasets used for all the experiments:** For experiment results in tables 3.1 and 3.2, we use the publicly available
LJSpeech dataset. For section 3.3, we use a subset of the MusicNet dataset (Thickstun et al., 2016) similar to Mor et al.
(2018). For the VQ-VAE experiment, we use the piano dataset provided by Dieleman et al. (2018).

**State-of-the-art claims for spectrogram-to-waveform inversion:** The authors would like to clarify that MelGAN is
a state-of-the-art *non-autoregressive* method for spectrogram-to-waveform inversion *trained from scratch* (does not
require KL-distillation from a teacher autoregressive model). Since this definition is quite narrow, we will clarify in the
final version that MelGAN is a *high quality* (instead of state-of-the-art) spectrogram-to-waveform inversion method.
Admittedly, autoregressive methods such as WaveNet and WaveRNN are slightly better at this task, but we believe
future work along this direction will close the gap.

**State-of-the-art claims for text-to-speech:** Furthermore, we will remove the state-of-the-art TTS claim made in
line 87 in the final version. This claim was initially made since MelGAN paired with text2mel shows the highest
reported MOS (of 3.88) on the publicly available LJSpeech dataset, beating Tacotron2 paired with WaveGlow (at 3.71).
The MOS of ground truth audio in this dataset is 4.72. We did not explicitly compare with Tacotron2 paired with
WaveNet since Prenger et al. (2019) show that WaveGlow performs similar to WaveNet in ground truth mel-spectrogram
reconstruction. However, we agree that a more direct comparison in the TTS setting is necessary to substantiate our
claim. Note that the MOS scores reported in the original Tacotron2 paper cannot be reproduced / compared with due to
the unavailability of the dataset or the original code.

**Discrepancy in MOS scores between Table 3 and Table 2:** The scores for the ablation study in Table 2 specifically
compares the importance of different components of the final MelGAN model. For this purpose, we only trained each
model for 400,000 iterations (1/6th the time required for the final converged model used in Table 3, which is trained for
2.4 million iterations). This is the reason for the discrepancy in MOS scores in the two tables.

**Updates for the final version:** The authors will add additional ground truth spectrogram-to-waveform inversion MOS
results for MelGAN compared with WaveNet, WaveGlow and original audio, as well as a stronger Text2Mel + WaveNet
baseline for TTS. We will refrain from claiming state-of-the-art unless substantiated by these tables.

**R1** *: claiming "autoregressive models can be readily replaced with MelGAN decoder" (line 89, line 228) without*
*necessary experiments*

We would like to clarify that this statement was not meant to convey that the perceptual quality of MelGAN decoder is
equivalent or better than autoregressive decoders in general. This statement was only meant to express the fact that the
MelGAN decoder is successfully shown to work in 3 different experimental setups that traditionally use autoregressive
decoders, such as : 1.) inverting mel-spectrograms to the corresponding acoustic waveform, 2.) mapping discrete latents
produced by a discrete variational auto-encoder to its corresponding observed waveform, 3.) mapping latent codes
produced by a Universal Music Translation Network to the corresponding raw waveform. We believe that this evidence
is sufficient to claim that MelGAN decoder is robust enough to replace autoregressive models for strongly conditional
waveform synthesis. We will update the paper to better reflect our intention.

In addition, for quantitative analysis of the performance of MelGAN on unseen speakers (without finetuning), we report
MOS scores on ground truth mel-spectrogram inversion on the VCTK dataset. We believe that this will serve as a good
task to test generalization for future research along this direction. For the sake of brevity, the results are as follows:
Original ($4.19 \pm 0.083$), MelGAN ($3.49 \pm 0.098$), Grifin Lim ($1.72 \pm 0.07$). Note that Griffin Lim is rated poorly as
there was no additional noisy baseline to anchor the scores causing a stark contrast between Griffin Lim and MelGAN.

**R1** *: [...] it's interesting to know how much benefit GAN brings in this work. A baseline to compare with is to train only*
*a Generator model with MSE loss (or other simple loss), without using Discriminator.*

This was an obvious first experiment that we tried. The model completely fails to capture the structure of the acoustic
waveform resulting in pure silence.

**R3** *: For the comparison in Table 1, it isn't clear at all whether the same hardware was used [...]*

Thanks for the feedback. Yes, the exact same hardware and computing specifications were used to compare all the
models. We will clarify this in the footnote.

[Meta-Review · NeurIPS 2019]

The paper describes a successful approach for non-autoregressive spectrogram inversion based on Generative Adversarial Networks. The reviewers noted that even though the results are not at the level of state-of-the-art, the paper addresses a difficult and timely problem, with a convincing experimental validation and ablation study. The rebuttal addressed the main concerns of the reviewers; the authors should nonetheless make sure to address other concerns in the camera-ready version.